# Genetic Basis Identification of a *NLR* Gene, *TaRGA5-like*, That Confers Partial Powdery Mildew Resistance in Wheat SJ106

**DOI:** 10.3390/ijms25126603

**Published:** 2024-06-15

**Authors:** Xiaoying Liu, Chenxiao Yang, Siqi Wu, Huixuan Dong, Guangyu Wang, Xinyue Han, Baoli Fan, Yuntao Shang, Chen Dang, Chaojie Xie, Zhenying Wang

**Affiliations:** 1Tianjin Key Laboratory of Animal and Plant Resistance, College of Life Sciences, Tianjin Normal University, Tianjin 300387, China; skylxy@tjnu.edu.cn (X.L.); yangchenxiao0306@163.com (C.Y.); wusiqi_2024@163.com (S.W.); donghuixuan2024@163.com (H.D.); wgyu2024@163.com (G.W.); 15075884694@163.com (X.H.); skyfbl@tjnu.edu.cn (B.F.); 2Tianjin Key Laboratory of Water Resources and Environment, Tianjin Normal University, Tianjin 300387, China; shangyuntao@126.com; 3Key Laboratory of Crop Heterosis and Utilization (MOE), State Key Laboratory for Agro-Biotechnology, Beijing Key Laboratory of Crop Genetic Improvement, China Agricultural University, Beijing 100193, China; dangchensa@163.com (C.D.); xiecj127@126.com (C.X.)

**Keywords:** *Triticum aestivum* L., powdery mildew, *PmSJ106 locus*, *TaRGA5-like*, partial resistance

## Abstract

Wheat powdery mildew is an important fungal disease that seriously jeopardizes wheat production, which poses a serious threat to food safety. SJ106 is a high-quality, disease-resistant spring wheat variety; this disease resistance is derived from Wheat-wheatgrass 33. In this study, the powdery mildew resistance genes in SJ106 were located at the end of chromosome 6DS, a new disease resistance locus tentatively named *PmSJ106 locus*. This interval was composed of a nucleotide-binding leucine-rich repeat (NLR) gene cluster containing 19 NLR genes. Five NLRs were tandem duplicated genes, and one of them (a coiled coil domain–nucleotide binding site–leucine-rich repeat (CC-NBS-LRR; CNL) type gene, *TaRGA5-like*) expressed 69–836-fold in SJ106 compared with the susceptible control. The genome DNA and cDNA sequences of TaRGA5-like were amplified from SJ106, which contain several nucleotide polymorphisms in LRR regions compared with susceptible individuals and Chinese Spring. Overexpression of *TaRGA5-like* significantly increased resistance to powdery mildew in susceptible receptor wheat Jinqiang5. However, Virus induced gene silence (VIGS) of *TaRGA5-like* resulted in only a small decrease of SJ106 in disease resistance, presumably compensated by other *NLR* duplicated genes. The results suggested that *TaRGA5-like* confers partial powdery mildew resistance in SJ106. As a member of the *PmSJ106 locus*, *TaRGA5-like* functioned together with other *NLR* duplicated genes to improve wheat resistance to powdery mildew. Wheat variety SJ106 would become a novel and potentially valuable germplasm for powdery mildew resistance.

## 1. Introduction

Wheat powdery mildew is an important fungal disease that seriously jeopardizes wheat production. Over the years, the wheat yield loss caused by *Blumeria graminis* f. sp. *tritici* (*Bgt*) generally reaches 10–15% and up to 50% in severe cases [1]. Selection and promotion of varieties resistant to powdery mildew is an important objective of wheat genetic improvement and always the most cost-effective measure to ensure food security and maintain green and safe agriculture.

In order to broaden and enrich the source of wheat powdery mildew resistance, scientists have been continuously discovering powdery mildew resistance genes from wheat commercial varieties, local varieties, wild relatives, and distantly related species. So far, 69 powdery mildew resistance (*R*) genes have been formally named (*Pm1*–*Pm69*), and there are also many other temporarily named powdery mildew *R* genes or alleles, of which only 17 *R* genes have been cloned. *Pm3b*/*Pm8*/*Pm17*, *Pm2a*, *Pm21*/*Pm12*, *Pm60*/*MlIW172*/*MlWE18*, *Pm5e*, *Pm41*, *Pm1a*, and *Pm69* encode nucleotide-binding leucine-rich repeat (NLR) immune receptor proteins [2,3,4,5,6,7,8,9,10,11,12,13,14]. *Pm24* and *WTK4* encode tandem kinase proteins [15,16]. *Pm4* encodes a chimeric protein recognized as a serine/threonine kinase, which contains multiple C2 structural domains and transmembrane regions [17]. *Pm13* and *Pm36* encode kinase fusion proteins [18,19]. *Pm38* (*Lr34*/*Yr18*/*Sr57*, ABC transporter) and *Pm46* (*Lr67*/*Yr46*/*Sr55*, hexose transporter) show a broad spectrum of resistance to powdery mildew and rust with broad-spectrum adult plant resistance [20,21]. Up to 70% of the major, dominant, and race-specific *R* genes in barley and wheat are encoded by typical NLR immune receptors. Due to the frequent mutation of wheat powdery mildew, the emergence and prevalence of new highly virulent pathogens have led to the rapid loss of powdery mildew resistance in selected and popularized wheat varieties, posing a serious challenge to wheat breeding and production [22]. For example, *Pm1*, *Pm2*, *Pm3*, *Pm4*, *Pm5*, and *Pm8* are losing their resistance to *Bgt* [23,24,25]. Even the most effective gene, *Pm21*, derived from *Dasypyrum villosum*, is also facing the risk of losing powdery mildew resistance [26]. Therefore, the identification and cloning of new *R* genes and their use in cultivated wheat germplasm could enrich the gene resource for resistance breeding.

SJ106 was the high-generation sister line of the wheat variety Jinqiang9, which bred from the 03 Yuan Yin-15/02K-53) F_6_ generation line as a male parent and (315//Wheat-wheatgrass 33//02K-53, 3/02K-53-5) F_4_ generation line as a female parent [27]. SJ106 is a high-quality, disease-resistant spring wheat line that is resistant to powdery mildew and moderately resistant to leaf rust and stripe rust. SJ106 showed strong resistance to powdery mildew during whole growth, suggesting that it is a potential powdery mildew-resistant gene donor. Therefore, it is of great significance to carry out the genetic location, cloning, and functional analysis of the powdery mildew resistance gene in SJ106 and transfer it to wheat varieties with excellent agronomic traits for wheat breeding.

In this study, we used Chinese Spring and SJ106 to construct F_2_ and F_2:3_ populations and located a new disease resistance locus. *PmSJ106 locus* was located at the end of chromosome 6DS by bulked segregant RNA-Seq (BSR-seq) and simple sequence repeat (SSR). In this locus, nineteen *NLR* genes were located, five *NLRs* were duplicated gene clusters, and one of them (coiled coil domain–nucleotide binding site–leucine-rich repeat (CC-NBS-LRR; CNL), *TaRGA5-like*) within this cluster was expressed 69–836-fold in SJ106 compared with the susceptible control. Over-expression of *TaRGA5-like* significantly increased resistance to powdery mildew in susceptible receptor wheat. However, VIGS silencing of *TaRGA5-like* resulted in only a small decrease in disease resistance in SJ106, presumably compensated by other duplicated *NLRs*. The above results suggested that the duplicated *NLRs* in *PmSJ106* gene clusters functions together in wheat resistance to powdery mildew infection. *TaRGA5-like*, as a member of this gene cluster, confers partial powdery mildew resistance in SJ106.

## 2. Results

### 2.1. Genetic Analysis of Powdery Mildew Resistance in SJ106

In order to investigate the disease resistance of SJ106, we constructed F_2_ and F_2:3_ populations using Chinese Spring and SJ106. After 10 days post inoculation (dpi) with *Bgt E09*, there were no visible powdery mildew spores on the surface of SJ106 leaves (infection types, IT = 0), which showed immunity to powdery mildew. In contrast, Chinese Spring wheat was highly susceptible to powdery mildew with a lesion area greater than 80%, and the IT grade was 4 (Figure 1). The F_1_ plants from the cross between Chinese Spring and SJ106 showed immunity or near-immunity to *Bgt E09* (IT = 0, 0;), which was consistent with the types of responses to *Bgt* infection in SJ106. The F_2_ plants showed different *Bgt E09* resistance (Figure 1). The segregation ratio of F_2_ individuals of Chinese Spring × SJ106 was close to 3:1. The segregation ratio of the F_2:3_ family lines was also consistent with 1:2:1 (Appendix A). Therefore, it was hypothesized that the seedling resistance of wheat SJ106 to *Bgt E09* was controlled by a dominance major gene locus, and it was tentatively named *PmSJ106 locus*.

### 2.2. SNP Calling and Confirmation of Candidate Interval

The total RNA of 40 F_2_-resistant individuals and 40 susceptible individuals were mixed respectively to construct the resistance/susceptibility pools and sequenced by BSR-seq. A total of 79,381,812 and 76,097,016 raw reads were retained after quality control. A total of 77,458,450 and 73,566,956 clean reads were obtained from the resistance/susceptibility pools, respectively. The percentages of clean reads with a Q30 were 90.34% and 91.74%, and the GC contents were 57.33% and 56.77%, respectively. A total of 65,018 high-quality differential SNPs were selected for subsequent SNP index analysis. Using the SNP index and 0.21 as the threshold, two candidate intervals were located on chromosome 6A (22.02–61.42 Mb) and 6D (7.7–56.02 Mb), respectively (Figure 2).

### 2.3. Mapping of Powdery Mildew Resistance Gene

In order to localize the *R* genes in the SJ106 chromosome, we used 1388 pairs of specific primers distributed on 21 wheat chromosomes to screen for specific markers in the F_2_ resistance/susceptibility pools of SJ106 and Chinese Spring, the resistance/susceptibility pools of the F_2:3_ family lines, and the F_2_ individuals. It was found that seven SSR markers (BARC183, BARC365, CFD49, TC93912, TC91050, TC66898, and BJ212775) were able to consistently amplify the polymorphic bands in parents, resistance/susceptibility pools, and resistant and susceptible individuals (Figure 3). A genetic linkage map was constructed based on the seven SSR markers. The powdery mildew resistance gene *PmSJ106* was located on chromosome 6DS and flanked by the markers BARC183 and TC93912, with genetic distances of 1.6 cM and 2.8 cM, respectively (Figure 4). The candidate region of the *PmSJ106 locus* was located between 4.90–8.97 Mb using the IWGSC Chinese Spring reference genome v1.1 (Figure 4). Combined with 2.2 results, the candidate region was located on chromosome 6DS, and the physical region was between 4.90–8.97 Mb. As no *Pm* genes were reported in this region, the *PmSJ106 locus* is likely to contain a novel *Pm*-resistant gene.

### 2.4. PmSJ106 Locus Region in Pan-Genome Is Enriched with Multiple NLR and Kinase Genes

Since BARC183 and TC93912 are molecular markers linked to disease resistance genes, the candidate genes in this interval were analyzed. To obtain a more exhaustive list of gene information, we analyzed the corresponding candidate intervals in several hexaploid wheats and durum wheat. The size of the *PmSJ106 locus* ranged between 4.07 to 7.44 Mb across the pan-genome, and there were 117 to 482 genes within this interval. As reported, *Pm* genes were always NLRs, transporter, and kinase, such as *Pm21* and *Pm13*. We further annotated candidate genes in this region. Nineteen genes were predicted to encode NLRs, and forty encoded kinase proteins. Fifteen of forty kinases were annotated as LRR-receptor like protein kinases (Appendix A). 

To determine the genetic relationship of NLRs and kinases, we obtained the gene sequences of the Chinese Spring genome (V2.1) and align them using multiple sequence alignment tool. According to the previous reports, this resulted in the formation of two distinct clusters, of which group I contained seven of the nineteen up-regulated NLR genes (Figure 5A). The kinases formed three clusters, each containing a single up-regulated kinase gene (Figure 5B). The tandem duplicated analysis of NLRs within the candidate intervals showed that *TraesCS6D03G0026700*, *TraesCS6D03G0026800*, *TraesCS6D03G0027300*, *TraesCS6D03G0027400*, and *TraesCS6D03G0027500* were tandem duplicated genes. The on-line expression analysis results showed that *TraesCS6D03G0027500* expressed higher after *Bgt* infection and lower in susceptible CS (Appendix A).

### 2.5. Cloning of the Candidate Powdery Mildew Resistance Gene in SJ106

The full length of the *TraesCS6D03G0027500* homologous gene was amplified in SJ106, which had a DNA length of 4254 bp, an open reading frame of 3138 bp, and encoded 1045 amino acid residues (Figure 6). There were 17 different amino acids (Appendix A). The protein functional structure domains were predicted by NCBI/CD-Search, and it was found to have a typical N-terminal RxCC like, nucleotide-binding-site (NBS), and leucine-rich repeat (LRR) domain belonging to the CNL protein family, sharing high homology with RGA5-like (XP_044417598.1) and termed as *TaRGA5-like*. The homologous genes of *TraesCS6D03G0027500* were identified in other wheat varieties (Appendix A). The sequences alignment result showed that they share high homology, and there were some nucleotide differences in LRR regions (Appendix A). The comparison of LRR region sequences showed that resistance individuals were almost consistent, while the susceptible individuals were the same as the sequence in Chinese Spring genome (Appendix A).

### 2.6. Expression Analysis of TaRGA5-like

The expression patterns in Chinese Spring and SJ106 were analyzed after *Bgt* infection (Figure 7). In resistant wheat SJ106, expression levels began to be significantly up-regulated at 12 hpi, and the highest expression was reached at 5 dpi, although it began to be down-regulated later. The expression level remains significantly higher than the non-infected control. In Chinese Spring, gene expression levels were very low and began to decrease after infection. The expression levels of the disease-resistant SJ106 were 69–836 times higher than that of Chinese Spring (Figure 7).

### 2.7. Subcellular Localization of TaRGA5-like

Tobacco leaves were infiltrated with Agrobacterium tumefaciens containing a binary vector carrying *TaRGA5-like-GFP* to transiently express and observe TaRGA5-like-GFP with a fluorescence microscopy. 35S: GFP distributed the whole cells. TaRGA5-like-GFP was localized in the cell membrane (Figure 8). This indicated that TaRGA5-like was localized in the cell membrane.

### 2.8. VIGS of of TaRGA5-like

To further analyze the function of *TaRGA5-like*, we transiently silenced *TaRGA5-like* using virus-induced gene silencing (VIGS). RT-qPCR results showed that the expression level of *TaRGA5-like* was significantly down-regulated in *TaRGA5-like* silenced leaves (Appendix A). After inoculation with *Bgt* for 7d, 10d, and 14d, the phenotypes of leaves look similar. After microscopic observation, it was found that the average amounts and densities of *Bgt* spores on *TaRGA5-like* silenced leaves increased from 145 to 557 at 7 dpi and from 1908 to 3702 at 14 dpi (Appendix A). The amounts of spores on *TaRGA5-like* silenced leaves were higher than that in the controls, which indicated disease resistance reduction. The proportions of hypersensitive reactions (HR) in the leaves decreased from 12.46‰ to 8.40‰ at 7 dpi and from 28.13‰ to 20.56‰ at 14 dpi (Appendix A). The reduction of HR in *TaRGA5-like* silenced leaves indicated that a decreased disease resistance (Figure 9). The above results indicated that *TaRGA5-like* silencing reduced powdery mildew resistance in SJ106 to some extent.

### 2.9. Over-Expression of of TaRGA5-like in Susceptible Wheat

We constructed the ubi:*TaRGA5-like* vector, which was transferred into susceptible spring wheat Jinqiang5. After PCR analysis, two transgenic wheat plants were selected (Appendix A). The transgenic wheat was cultured to the one-leaf stage and inoculated with Bgt for 7 d. The results showed that T_1_ transgenic plants were resistant to Bgt E09. The lesion area on the leaf surfaces of transgenic wheat was significantly lower than that of the control (Figure 10A). The lesion area decreased from nearly 60% to 19.45% and 26.69, respectively (Figure 10B, Appendix A). At 7 dpi, fewer secondary hyphae (SH) or beaded conidia (BC) grew on the TaRGA5-like OE leaves, whereas the control leaves were major covered by BC (Figure 11A). Statistical analysis showed that the percentage of successful penetration was lower in all OE wheat leaves than that in the controls (Figure 11B), whereas the macrocephalic appressoria and hypersensitive reactions rates were the opposite (Figure 11C,D). The OE results indicated that over-expression of TaRGA5-like in Jinqiang 5 improved wheat resistance.

## 3. Discussion

Cultivating resistant varieties is an economical, effective, and environmentally friendly measure. But due to the rapid evolution of pathogens, resistance in varieties is often overcome, resulting in disease re-emergence [22,28]. This requires new disease-resistant varieties with durable, broad-spectrum resistance characteristics. Most *R* genes encode typical immune receptors. Although individual *R* gene resistance is easily overcome by targeted selection of pathogenic strains, *R*-gene polymerization can result in durable, broad-spectrum disease resistance. Further research on NLRs and the integrating structural domains (IDs) that act as decoys for pathogen effectors is still very important. The core idea of *R*-gene polymerization is that, as the number of genes introduced, transformed, or cultivated into the same crop variety increases, the difficulty for pathogens to evolve corresponding multi-gene polymerization resistance virulence becomes greater. For example, different degrees of broad-spectrum resistance have been obtained by transgenic technology by directly transferring the wheat adult-phase persistent multi-resistance genes *Lr34*/*Yr18*/*Sr57*/*Pm38* into several crops, including durum wheat, barley, and rice [29,30,31]. Transformation of wheat *NLR* genes *Sr22*, *Sr33*, *Sr35*, and *Sr45* into the susceptible barley variety Golden Promise exhibited small species-specific resistance to barley rust without any significant agronomic effects [32].

SJ106 is a high-quality spring wheat variety, resistant to powdery mildew and moderately resistant to leaf rust and stripe rust. Based on the hybridized combinations of SJ106, it was hypothesized that the source of resistance in SJ106 was from the disease-resistant wheat variety Wheat-wheatgrass 33. In this study, we constructed F_2_ and F_2:3_ genetic populations of Chinese Spring/SJ106. The powdery mildew-resistant gene in SJ106 was located at the end of chromosome 6DS (4.90–8.97 Mb) using SSR and BSR-seq. In previous reports, only one, *Pm45*, was identified on wheat chromosome 6D (105.5–160.4 Mb), which was far away from the gene locus we localized [28]. No *Pm* gene was reported in the localized interval, except for a *Sr42 locus* [33].

By comparing several hexaploid wheat genomes and durum wheat sequences, it was found that this interval contained 19 *NLR* gene loci, and the frequency of *NLR* distribution (19/4.07 Mb) was much higher than the average frequency of distribution of NLR loci in the wheat genome (2500/16,000 Mb), and it was a cluster of *NLR* genes. Gene duplication and diverse selection have resulted in a clustered distribution of NLRs in plant genomes. Pan-genomic variants, such as PAVs, are particularly associated with *NLRs* and other types of *R* genes. Usually, *NLRs* that regulate resistance were absent or functionally reduced in susceptibility genes [22]. The duplication analysis results of *NLRs* annotated in the Chinese Spring genome v2.1 showed that five NLR genes (*TraesCS6D03G0026700*, *TraesCS6D03G0026800*, *TraesCS6D03G0027300*, *TraesCS6D03G0027400*, and *TraesCS6D03G0027500*) were tandem duplicated genes with a homology of over 80%, suggesting similar functions. Therefore, the highly similar genes in a *PmSJ106* cluster make locating a causal NLR particularly challenging.

The on-line expression analysis results showed that *TraesCS6D03G0027500* expressed higher after *Bgt* infection. Further RT-qPCR results showed that the expression of *TaRGA5-like* (*TraesCS6D03G0027500*) was significantly higher in the disease-resistant SJ106 than in Chinese Spring. We amplified *TaRGA5-like* in SJ106 and found that the sequence differed significantly from the homologous sequences in Chinese Spring, Fielda, and Kenong, and the differences were mainly concentrated in the LRR region. The comparison of LRR region sequences in F_2_ individuals showed that resistance individuals were almost consistent, while the susceptible individuals were the same as the sequence in the Chinese Spring genome. Therefore, we speculated that the differences in LRR region led to the differences in the recognition of powdery mildew effector by TaRGA5-like, eventually resulting in resistance differences among wheat varieties.

To further analyze *TaRGA5-like* function, we constructed over-expression transgenic plants and VIGS transient gene silencing plants, respectively. Over-expression of *TaRGA5-like* significantly increased the resistance of susceptible recipient wheat to powdery mildew. However, the disease resistance of SJ106 only slightly decreased after VIGS of *TaRGA5-like*. We speculate that this may be due to the tandem duplicated NLRs of *TaRGA5-like*, such as *TraesCS6D03G0026700*, *TraesCS6D03G0026800*, *TraesCS6D03G0027300*, and *TraesCS6D03G0027400*, which compensated for the disease-resistant effect of *TaRGA5-like* and thus maintained the powdery mildew resistance in SJ106. Taken together, the *PmSJ106 locus* is a natural cluster of duplicated *R* genes formed by the polymerization of multiple disease-resistant *NLRs*, which work together to help SJ106 form a broad-spectrum and durable resistance in the whole growth period. The duplicated *NLRs* in the *PmSJ106 locus* had the same genetic background, and the disease-resistant *NLRs* complemented each other, achieving natural polymerization of *R* genes and avoiding the problem of disease resistance masking that may occur during the polymerization process of different genetic *R* genes. Wheat variety SJ106, which contained a natural *R* gene cluster, could become a novel and potentially valuable germplasm for powdery mildew resistance and contribute to the green development of wheat breeding.

## 4. Materials and Methods

### 4.1. Plant Material

SJ106 is a high-quality, disease-resistant spring wheat line bred from the 03 Yuan Yin-15/02K-53) F_6_ generation line as a male parent and (315//Wheat-wheatgrass 33//02K-53, 3/02K-53-5) F_4_ generation line as a female parent. The wheat cultivars Chinese Spring, susceptible to powdery mildew, were used as susceptible parents and crossed with SJ106 to obtain F_1_ hybrids, F_2_ populations, and F_2:3_ families, respectively. F_1_ hybrids, F_2_ populations, and F_2:3_ families were used for genetic analysis of the powdery mildew resistance in SJ106. All seeds were stored in our lab (Tianjin, China).

The disease-resistant parent (SJ106), the susceptible parent (Chinese Spring), and the individuals of Chinese Spring/SJ106 (10 F_1_, 285 F_2_, and 630 F_2:3_) were sterilized in a 75% alcohol solution and sown in 72-well seedling trays with 5 seeds per well, and one highly susceptible powdery mildew cultivar, Chinese Spring, was sown in every 4 holes to ensure the full inoculation of the experimental materials. After sowing, the seedling trays were placed in an artificial climate chamber (relative humidity of 70%, light intensity of 125 μmol m^−2^s^−1^, photo-period of 12 h, and temperatures of 22 °C in light and 16 °C in the dark), and planted for about 7 d. When the second leaf of the seedlings was fully expanded, the seedlings were inoculated by manually sweeping with fresh *Bgt* spores, and five seedlings were inoculated for each material. After about 10 days post inoculation (dpi), the infection type of the materials was investigated according to the grading standard of 0~4, and the experiment was repeated three times to analyze the characterization.

### 4.2. BSR-Seq and RNA-Seq Analysis

A total of 40 homozygous resistant plants and 40 homozygous susceptible plants were sown in the culture plates. When the second leaf was fully expanded, *Bgt E09* was inoculated on the wheat leaves. The middle part of the second leaves were taken at a length of about 4 cm, and the total RNA was extracted following the instructions (Tiangen, Beijing, China) and mixed in equal quantities to construct the resistant and susceptible pools, respectively. The total RNA of the resistant and susceptible pools, the resistant parent SJ106, and the susceptible parent Chinese Spring were sent to select differential single nucleotide polymorphism (SNP) and differential expressed genes (DEGs) by BSR-seq and RNA-seq analysis (Novogene, Tianjin, China).

### 4.3. SSR Analysis

Genome DNA was extracted from 40 homozygous resistant plants and 40 homozygous susceptible plants, respectively. The genome DNA of the wheat was extracted following the CTAB method, and their DNA was mixed in equal amounts to construct resistant and susceptible pools for SSR analysis. Based on the primer sequences published in the wheat database GrainGenes (https://wheat.pw.usda.gov/GG3/) (accessed on 12 October 2019), polymorphisms were screened between the resistant and susceptible parents (SJ106 and Chinese Spring) and between the resistant and susceptible pools using bulked segregant analysis (BSA). Individual-plant identification was performed in the F_2_ populations to preliminarily infer the chromosomal location of the disease-resistant genes.

### 4.4. Candidate Gene Prediction and Functional Analysis

The relevant sequences of the two nearest markers flanking the target gene *PmSJ106* were used to align with the reported wheat reference genome (Chinese Spring v1.0 and v2.1, Arinalrfor, Lancer, Landmark, Mace, Norin61, Stanley, Jagger, Julius, Zang1817, Fielder, Renan, Landmark, Norin61, Kenong9204, SY Mattis, and Kariega) to determine the intervals in which the genes are located (http://wheatomics.sdau.edu.cn/) (accessed on 16 February 2024) [34]. Based on the gene annotations of the related reference genome, the physical positions were determined to predict candidate genes, and *R* genes within the localized interval were selected. The phylogenetic trees of NLRs and kinases sequences were obtained from the Chinese Spring genome (V2.1) and align them using MEGA (v11.0.13) software, respectively. The alignment results were clustered according the previous reports [35]. The tandem duplicated events of the *NLRs* in the intervals were carried out using MCScanX [36]. The expression levels of *NLRs* were analyzed online (http://www.wheat-expression.com) (accessed on 26 February 2024). To determine the genetic relationship of NLRs and kinases, we obtained the gene sequences of the Chinese Spring genome (V2.1) and align them using multiple sequence alignment tool. 

### 4.5. Isolation and Sequence Analysis of TaRGA5-like in SJ106

The sequences were amplified with PrimeSTAR GXL DNA Polymerase (TAKARA, Dalian, China). The amplification primers were designed according to the *TraesCS6D03G0027500* sequence, and all primer sequences were listed in Appendix A.

### 4.6. Expression Analysis of TaRGA5-like

The expression analysis by real-time quantitative PCR (RT-qPCR) was performed as described previously [37]. The actin gene was used as an internal control.

### 4.7. Subcellular Localization of TaRGA5-like

The experiment was performed as described previously [30]. The recombinant and control vectors (pCAMBIA1302:*TaRGA5-like* cDNA-*GFP* and pCAMBIA1302:*GFP*) were injected into tobacco epidermal cells, respectively, and observed after 72 h. The pictures were taken with fluorescence microscopy (80i, NIKON, Tokoyo, Japan).

### 4.8. Function Analysis of TaRGA5-like by VIGS

Barley stripe mosaic virus (BSMV) was used to silence *TaRGA5-like*. Three non-conserved regions of *TaRGA5-like* were chosen and inserted into BSMVγvector for VIGS analysis. BSMVα, BSMVβ, and BSMVγ were mixed equally and injected into wheat leaves. The related vectors and silenced approach were carried out as described previously [36]. Morphology and incompatibility reactions of powdery mildew spores were observed, and the number of appressoria, normal, and deformed spores were counted. The calculation method of HR was performed as described previously [38]. The photographs were taken of the phenotypes of the VIGS wheat leaves.

### 4.9. Function Analysis of TaRGA5-like by OE

The recombinant vector pTCK303:*TaRGA5-like* (ubi: *TaRGA5-like*) was transformed into the susceptible wheat variety Jinqiang5 using *Agrobacterium tumefacien* EHA105. The experimental methods were carried out following the previously published articles [36]. Two independent transgenic T_1_ seedlings (50 mg∙L^−1^ hygromycin) were identified by genomic PCR and qRT-PCR analysis. The infection types of two independent T_1_ transgenic lines were evaluated. Photographs were taken of the phenotypes of OE wheat leaves.

## 5. Conclusions

In this study, a new powdery mildew locus, *PmSJ106 locus*, was located on chromosome 6DS. In the locus, five *NLRs* were tandem duplicate genes, and the *TaRGA5-like* of them expressed significantly higher in SJ106. OE and VIGS results indicated that *TaRGA5-like* only confers partial *Bgt* resistance. We inferred that the function of *TaRGA5-like* could be compensated by other *NLR* duplicated genes after VIGS. In summary, we speculated that the *NLRs* in *PmSJ106 locus* functioned together to improve wheat resistance to powdery mildew as a genes cluster.

## Figures and Tables

**Figure 1 ijms-25-06603-f001:**
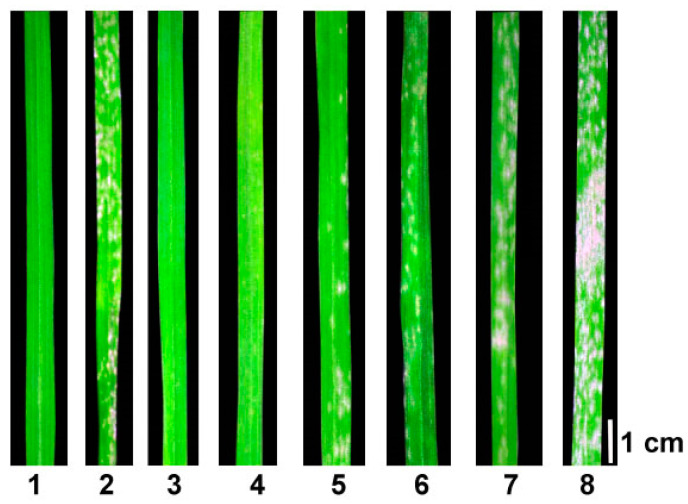
The seedling response of F_2_ populations and its parents to *Bgt E09*. 1: SJ106 (IT = 0), 2: Chinese Spring (IT = 4), 3: F_2_ individuals (IT = 0), 4: F_2_ individuals (IT = 0), 5: F_2_ individuals (IT = 1), 6: F_2_ individuals (IT = 3), 7: F_2_ individuals (IT = 3), 8: F_2_ individuals (IT = 4). Seedling IT values of 0–2 indicate resistance and 3–4 susceptibility.

**Figure 2 ijms-25-06603-f002:**
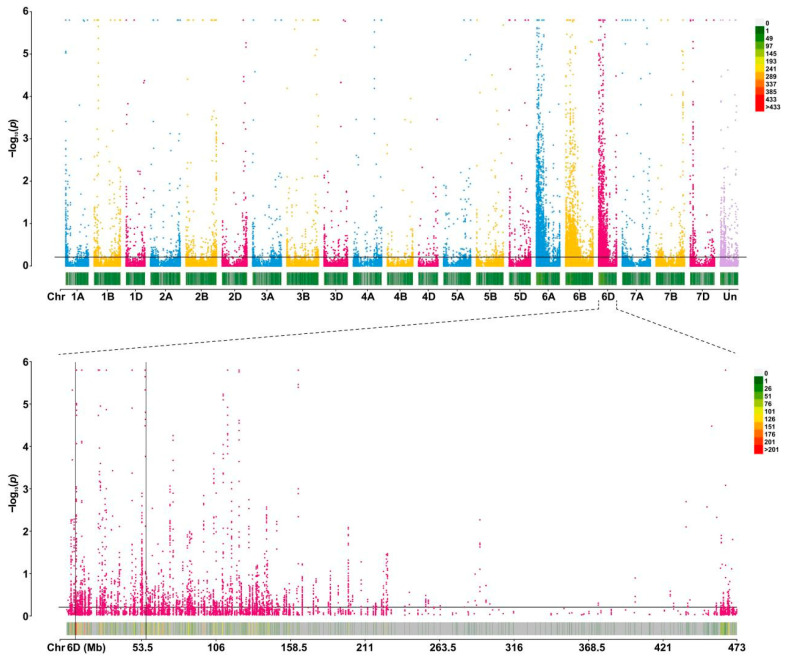
Differential SNPs analysis in SJ106.

**Figure 3 ijms-25-06603-f003:**
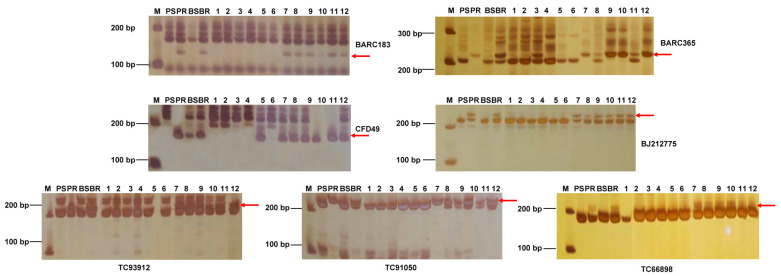
PCR-amplification with SSR markers in parents, resistant pool, and F_2_ individuals. M: DL 5000 Marker, PS: susceptible parent China Spring, PR: resistant parent, BS: susceptible pool, BR: disease-resistant pool, 1–6: susceptible individual plants; 7–12: disease-resistant individual plants. Red arrow pointed to the sepecific PCR-amplification with SSR marker in disease resistant samples.

**Figure 4 ijms-25-06603-f004:**
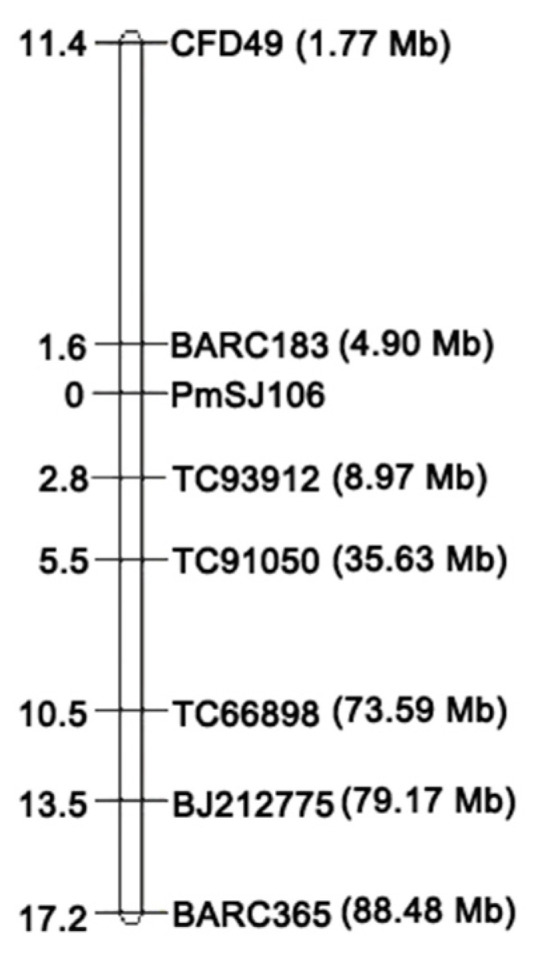
Genetic linkage map and physical location map of *PmSJ106 locus*.

**Figure 5 ijms-25-06603-f005:**
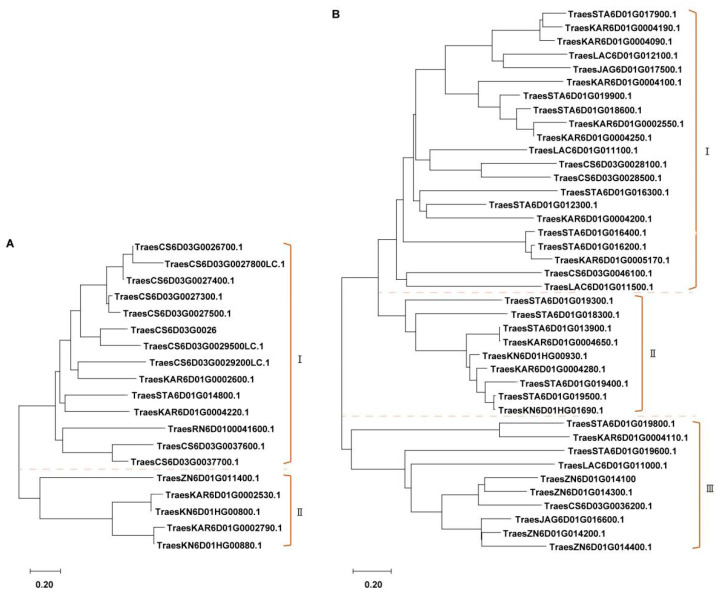
Phylogenetic analysis of NLRs and kinases. (**A**) Phylogenetic tree of NLRs from multiple wheat genome. (**B**) Phylogenetic tree of kinases from multiple wheat genome. The NLRs and kinases were clustered according to the homology of their amino acid sequences.

**Figure 6 ijms-25-06603-f006:**
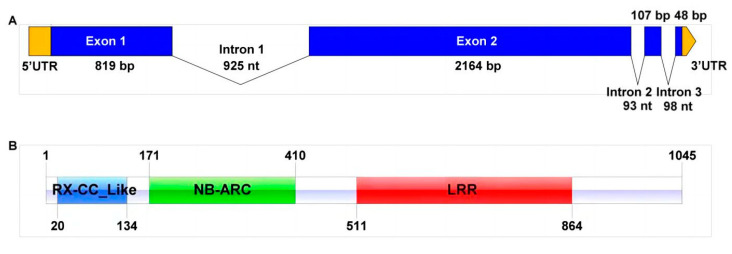
Structure characteristic analysis of TaRGA5-like. (**A**) Distribution of exons and introns. (**B**) Predicted structure of TaRGA5-like. Yellow: untranslated region (UTR); blue: exon; light blue: RX-CC_Like domain; green: nucleotide-binding adaptor shared by Apaf1, certain R genes and CED4 (NB-ARC) domain; red: leucine-rich-repeat (LRR) domain.

**Figure 7 ijms-25-06603-f007:**
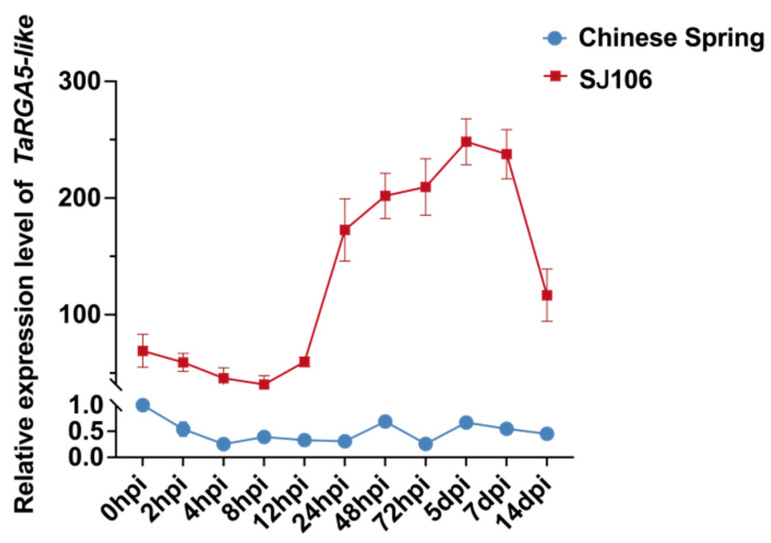
Expression analysis of *TaRGA5-like* after *Bgt* infection.

**Figure 8 ijms-25-06603-f008:**
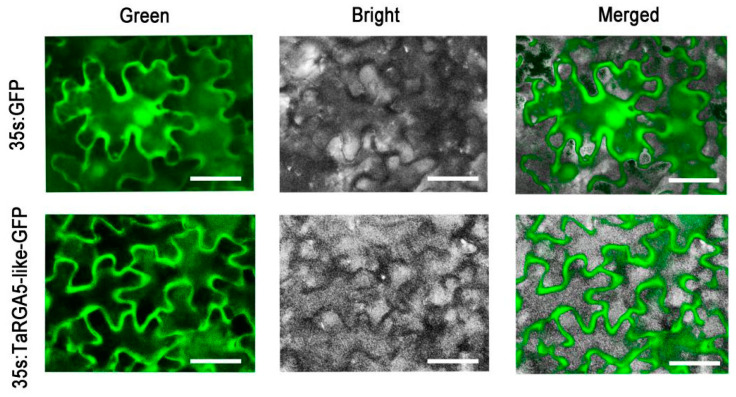
Subcellular location of TaRGA5-like fused with GFP in tobacco. 35S: GFP was GFP vector control. GFP emitted green fluorescence. Bar = 30 μm.

**Figure 9 ijms-25-06603-f009:**
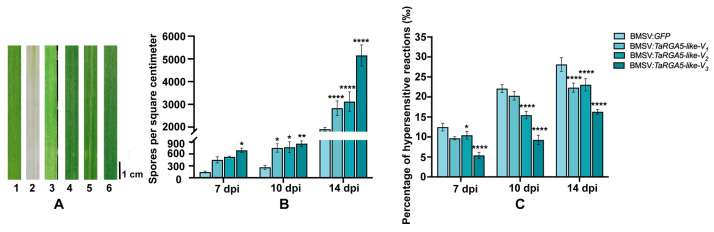
Silencing of *TaRGA5-like* reduces wheat resistance against *Bgt E09*. (**A**) Phenotype of BSMV silenced leaves at 14 dpi. 1: Control: SJ106, 2: BSMV:*PDS*: SJ106 infected with BSMV:*PDS*, 3: BSMV:*GFP*: SJ106 infected with BSMV:*GFP*, 4: BSMV:*TaRGA5-like-V1*: SJ106 infected with BSMV:*TaRGA5-like-V1*, 5: BSMV:*TaRGA5-like-V2*: SJ106 infected with BSMV:*TaRGA5-like-V2* and 6: BSMV:*TaRGA5-like-V3*: SJ106 infected with BSMV:*TaRGA5-like-V3*. (**B**) Spores per square centimeter. (**C**) Percentage of hypersensitive reactions in *TaRGA5-like* silenced SJ106 and control leaves. (* *p* < 0.05, ** *p* < 0.01, **** *p* < 0.0001).

**Figure 10 ijms-25-06603-f010:**
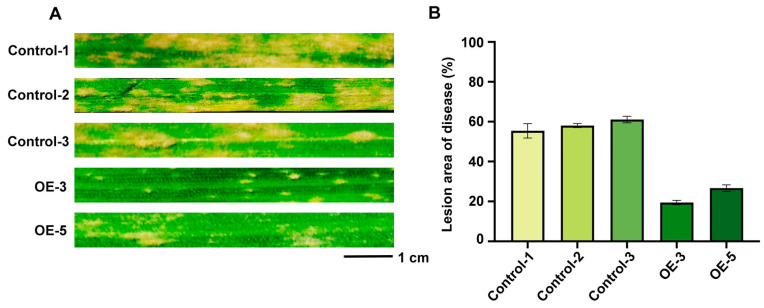
Over-expression of *TaRGA5-like* in susceptible wheat variety Jinqiang5. (**A**) Phenotype of *TaRGA5-like* OE and control wheat leaves inoculated with *Bgt* E09 for 7 dpi. (**B**) Lesion area of OE and control wheat leaves. Control-1: Jinqiang5; Control-2: Jinqiang5 transformed with *Agrobacterium tumefacien* EHA105; Control-3: Jinqiang5 transformed with *Agrobacterium tumefacien* EHA105 containing pTCK303 vector; OE-3, 5: transgenic Jinqiang5 wheat containing the recombinant vector pTCK303:*TaRGA5-like*.

**Figure 11 ijms-25-06603-f011:**
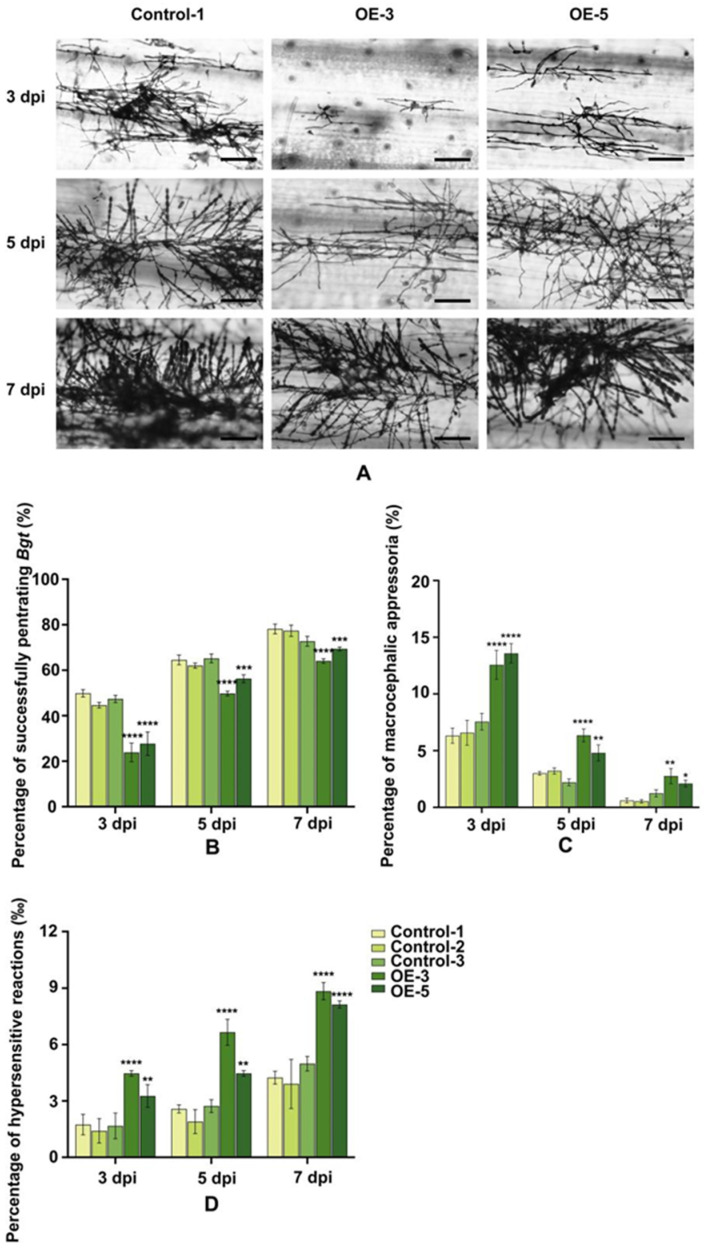
Infection parameters of SJ106 leaves at 7 dpi. (**A**) Spores phenotype of *TaRGA5-like* OE and control wheat leaves inoculated with *Bgt* E09 for 7 dpi. (**B**) Proportion of successful penetration by *Bgt*. (**C**) Percentage of macrocephalic appressoria. (**D**) Percentage of hypersensitive reactions in *TaRGA5-like* OE leaves and control leaves. (* *p* < 0.05, ** *p* < 0.01, *** *p* < 0.001, **** *p* < 0.0001). Bar = 100 μm.

## Data Availability

Data are contained within the article or Appendix A.

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
