# Peer review of "Genetic Basis Identification of a NLR Gene, TaRGA5-like, That Confers Partial Powdery Mildew Resistance in Wheat SJ106"

_ijms, 2024, doi:10.3390/ijms25126603_

Round 1
Reviewer 1 Report
Comments and Suggestions for Authors
Liu et al identified a new powdery mildew resistance gene locus, tentatively named PmSJ106 locus, in the chromosome 6DS of the high-quality spring wheat variant SJ106. They found that this interval clustered with 19 NLR genes, of which 5 NLRs genes were tandem repeated, and they further pursued one of them, TaRGA5-like, a CNL type gene. TaRGA5-like is highly expressed in SJ106 comparing with the susceptible counterpart Chinese Spring upon pathogen challenge. They found several nucleotide polymorphisms in LRR regions comparing with Chinese Spring and other susceptible individuals. They made transgenic wheat plants to over-express TaRGA5-like and VIGS-silenced wheat plants. The pathogenicity assays with powdery mildew were conducted to assess their resistance. Overall the manuscript is scientifically sound, however, lack of precise description of the results and what they conducted leads to confusion to some extent. I suggested that the author edit the manuscript throughout the text and all figure legends to make sure they clearly described.
Other comments follow and many others are not mentioned here.
Figure 1. The authors should indicate the type of responses: immune, intermediate, highly susceptible in the representative pictures equivalent to IT 0-4 for F2 population in both figure and figure legends. I suggest the order of leaf pictures: SJ106, Chinese Spring, then F2 individuals. The pictures of leaf symptoms look quite blurry.
Figure 5. Not be able to read. Higher resolution figures needed.
Figure 7. Inconsistency of labelling. You either use Chinese Spring or CS or SJ106 through the text. Is S Jian 106 SJ106? Remove ‘Different colour represents different wheat. Red: disease resistant near-isogenic line SJ106, blue: disease susceptible variety, Chinese Spring.’ since you already show with colored legends.
Figure 8. Bar=how long?
Figure 9. hard to read. They are all blurry.
Figure S1-S5 to combine in one single PDF file with figure legend.
Figure 10 and S5B, what are those controls? What is the mock control? Plant inoculated with water but why symptom showed? I don’t see any mock here, probably, I assume the mock is the non-transgenic susceptible plants inoculated with Bgt E09? pTCK303-TaRGA5-like? EHA101? Need to describe clearly what they are in order to let reader understand.
Figure 11. too small and too blurry.
Table S2, add an additional column to show the actual ratio. Seg is not appropriate, you should use ‘intermediate’ in the context of HR and HS.
I suggest use Chinese spring instead of CS; Bgt should appear as itlaticised through the text, in my opinion.
Line 49. …(NLR) immune receptor proteins.
Line 60. …are losing or have lost resistance to Bgt (what kind of strains or races?)?
Line 60. The currently most effective or the all times most effective?
Line 65. What is the origin of ‘Wheat-wheatgrass’ or ‘315//Wheat-wheatgrass 33//02K-53, 3/02K-53-5’? Nothing has been described or any literatures has been cited for this breed?
Line 66-68. The disease resistance assessment carried out in 2019 in the Institute of Plant Protection, Tianjin Academy of Agricultural Sciences, showed that it was resistant to powdery mildew,….
Line 69. …strong resistance?
Line 71. What does localization mean here? Genetic location?
Line 70-74. Please rephrase the sentences.
Line 75. ….we found that a gene cluster containing 19 NLR genes was located at the end of SJ106 chromosome 6DS,….
Line 77. Interval? Your mean ‘locus’?
Line 78. CNL has not been defined. I assume ‘Coiled coil domain-nucleotide binding site-leucine rich repeat (CC-NBS-LRR; CNL)’
Line 78. ..repeated genes. Duplicated genes?
Line 88. …days post inoculation (dpi). Please describe the inoculation with what strain. Bgt E09, I guess. Also your statement ‘no visible powdery mildew spores or hypersensitive reactions on the surface of SJ106 leaves…’ apparently conflict with your data shown in Table S2, in which showed 40 HR, if HR stands for hypersensitive reactions.
Line 89. What is ‘IT’, define it.
Line 90-91. In contrast, CS wheat was highly susceptible to powdery mildew with lesion area greater than 80%, and the IT grade was 4.
Line 93. the infection type in SJ106? You mean ‘the types of responses to Bgt infection?
Line 93-94. Get rid of ‘Thus, the powdery mildew resistance in SJ106 was controlled by the major gene locus.’
Line 104. What is BSR-sequencing? define it, please.
Line 111. What is ‘the results of part 3.2,..’, nothing about this is mentioned in the test.
Line 116. SJ106 chromosome?
Line 119. Spell out ‘SSR’
Line 120. amplify consistently the polymer…
Line 122. A genetic linkage map..?
Line 125. …between 4.90-8.97.., Is this a coordinate in genome?
Line 125-128, rephrase the sentences. It looks like redundancy.
Line 128. Pm resistant genes?
Line 131. PCR-Amplification with SSR markers in parents, resistant pool and F2 individuals.
Line 133. …individual plants.
Line 141. 59, Fifty-nine..
Line 142-143. ‘..were predicted to encode putative R proteins where 19 encoded NLRs and the remaining 40 encoded kinase proteins.’. I think there is a confusion here. Historically R-proteins were NLRs, however, you need to make sure not all kinase proteins are categorized as R-proteins. It would be better if you mention these as resistance-associated proteins?
Line 143. Fifteen of forty…
Line 145-146. …we compared the gene sequences of the Chinese Spring genome (V2.1).
Line 161. …a typical..
Line 163. TaRGA5-like? Where is it originated? Should have a citation.
Line 164-166. It is the coded protein sequences that are more important. Why no show a protein alignment?
Line 186. Agrobacterium tumefaciens; binary vector?
Line 186-189. Tobacco leaves were infiltrated with Agrobacterium tumefaciens containing a binary vector carrying TaRGA5-like-GFP to transiently express and observe TaRGA5-like-GFP with a confocal fluorescence microscopy.
Line 189. Is that a confocal fluorescence microscopy?
Line 192. 35S: GFP, GFP vector control.
Line 193. Get rid of ‘Green fluorescent protein (GFP) emitted green’. All color shown technically are pseudo color.
Line 196 and in Material and Methods. VIGS is not clearly described. What VIGS vector is used here?
Line 200-202. The statement ‘These results indicated that TaRGA5-like silencing reduced powdery mildew resistance in SJ106 to some extent, but the decrease was not significant.’ conflict with the figure, which told the VIGS-silenced plants were significantly different. Fix it.
Line 210. Remove ‘Meanwhile,’
Line 231. Are ‘Cultivating and planting’ redundant?
Line 236. What does ‘polymerization of multiple R genes’ mean? Diversification?
Line 253. only one, Pm45, was identified …?
Line 259-260. Using the same unit ‘(19/4.07 Mb) vs (2,500/16 Gb)’.
Line 394. formal analysis? Are there any informal analysis?
Comments on the Quality of English LanguageQuality of English needs to be improved. Major revision is required for grammar and precise description.
Author Response
Dear Reviewer,
Thanks very much for taking your time to review this manuscript. I really appreciate all your comments and suggestions! Please find my itemized responses in below and my revisions/corrections in the re-submitted files.
Sincerely,
Zhenying Wang
Appended to this letter is our point-by-point response to the comments raised by the reviewers. All changes in the text are highlighted in different colors.
Figure 1. The authors should indicate the type of responses: immune, intermediate, highly susceptible in the representative pictures equivalent to IT 0-4 for F2 population in both figure and figure legends. I suggest the order of leaf pictures: SJ106, Chinese Spring, then F2 individuals. The pictures of leaf symptoms look quite blurry.
Answer: Thank you for the reviewer’s suggestion, we have modified the figure 1 and its related description. The changes were highlighted in red color in part 2.1.
Figure 5. Not be able to read. Higher resolution figures needed.
Answer: We have modified the figure 5, and improve the resolution.
Figure 7. Inconsistency of labelling. You either use Chinese Spring or CS or SJ106 through the text. Is S Jian 106 SJ106? Remove ‘Different colour represents different wheat. Red: disease resistant near-isogenic line SJ106, blue: disease susceptible variety, Chinese Spring.’ since you already show with colored legends.
Answer: According to the reviewer’s suggestion, we have modified Chinese Spring and SJ106 through the text, and updated the Figure 7. The changes were highlighted in red color.
Figure 8. Bar=how long?
Answer: We added the description in Figure 8 legends. Bar=30μm.
Figure 9. hard to read. They are all blurry.
Answer: We modified the contents as “After microscopic observation, it was found that the average amounts and densities of Bgt spores on TaRGA5-like silenced leaves increased from 145 to 557 at 7 dpi, and from 1908 to 3702 at 14 dpi (Table S4). The amounts of spores on TaRGA5-like silenced leaves were higher than that in the controls, which indicated the disease resistance reduction. The proportions of hypersensitive reactions (HR) in the leaves decreased from 12.46‰ to 8.40‰ at 7 dpi, from 28.13‰ to 20.56‰ at 14 dpi (Figure S6). The reduction of HR in TaRGA5-like silenced leaves indicated that a decreased disease resistance (Figure 9). The above results indicated that TaRGA5-like silencing reduced powdery mildew resistance in SJ106 to some extent. ” Figure 9 was redrew. The changes were highlighted in red color.
Figure S1-S5 to combine in one single PDF file with figure legend.
Answer: We added a Figure S6, and combined Figure S1-S6 in one single PDF including figure legend.
Figure 10 and S5B, what are those controls? What is the mock control? Plant inoculated with water but why symptom showed? I don’t see any mock here, probably, I assume the mock is the non-transgenic susceptible plants inoculated with Bgt E09? pTCK303-TaRGA5-like? EHA101? Need to describe clearly what they are in order to let reader understand.
Answer: We modified the Figure 10 and S6B (S5B before modification), and added the explanations of control, EHA105, pTCK303, and OE-3, 5. The detailed contents were “Control-1: Jinqiang5; Control-2: Jinqiang5 transformed with Agrobacterium tumefacien EHA105; Control-3: Jinqiang5 transformed with Agrobacterium tumefacien EHA105 containing pTCK303 vector; OE-3, 5: transgenic Jinqiang5 wheat containing the recombinant vector pTCK303:TaRGA5-like.” The changes were highlighted in red color.
Figure 11. too small and too blurry.
Answer: We modified the figure 11, and improve its resolution.
Table S2, add an additional column to show the actual ratio. Seg is not appropriate, you should use ‘intermediate’ in the context of HR and HS.
Answer: We added the actual ratio column in Table S2, and ‘Seg’ represented segregant population.
I suggest use Chinese spring instead of CS; Bgt should appear as itlaticised through the text, in my opinion.
Answer: We used Chinese spring instead of CS through the text, and modified Bgt in italic. The changes were highlighted in red color.
Line 49. …(NLR) immune receptor proteins.
Answer: Thank you for the reviewer’s suggestion, we added the “immune receptor” in the text. The changes were highlighted in red color.
Line 60. …are losing or have lost resistance to Bgt (what kind of strains or races?)?
Answer: As it was mixed Bgt. We modified the description. The changes were highlighted in red color.
Line 60. The currently most effective or the all times most effective?
Answer: We modified the description in Line 40. The changes were highlighted in red color.
Line 65. What is the origin of ‘Wheat-wheatgrass’ or ‘315//Wheat-wheatgrass 33//02K-53, 3/02K-53-5’? Nothing has been described or any literatures has been cited for this breed?
Answer: SJ106 was the high-generation sister line of wheat variety Jinqiang9. We added the explanation and a new reference in Line 67. The changes were highlighted in red color.
Line 66-68. The disease resistance assessment carried out in 2019 in the Institute of Plant Protection, Tianjin Academy of Agricultural Sciences, showed that it was resistant to powdery mildew,….
Answer: Thank you for the reviewer’s modification, we revised it. The changes were highlighted in red color.
Line 69. …strong resistance?
Answer: We rephrased Line 69. The changes were highlighted in red color.
Line 71. What does localization mean here? Genetic location?
Answer: We have used genetic location in stead of localization. The changes were highlighted in red color.
Line 70-74. Please rephrase the sentences.
Answer: We rephrased the sentences in Line 70-74.
Line 75. ….we found that a gene cluster containing 19 NLR genes was located at the end of SJ106 chromosome 6DS,….
Answer: Thank you for the reviewer’s modification, we revised it. The changes were highlighted in red color.
Line 77. Interval? Your mean ‘locus’?
Answer: We used ‘locus’ in stead of ‘interval’. The changes were highlighted in red color.
Line 78. CNL has not been defined. I assume ‘Coiled coil domain-nucleotide binding site-leucine rich repeat (CC-NBS-LRR; CNL)’
Answer: We modified as ‘Coiled coil domain-nucleotide binding site-leucine rich repeat (CC-NBS-LRR; CNL)’. The changes were highlighted in red color.
Line 78. ..repeated genes. Duplicated genes?
Answer: We used ‘duplicated’ in stead of ‘repeated’ in the text. The changes were highlighted in red color.
Line 88. …days post inoculation (dpi). Please describe the inoculation with what strain. Bgt E09, I guess. Also your statement ‘no visible powdery mildew spores or hypersensitive reactions on the surface of SJ106 leaves…’ apparently conflict with your data shown in Table S2, in which showed 40 HR, if HR stands for hypersensitive reactions.
Answer: We modified the description in Line 88. Because hypersensitive reactions couldn’t be observed in wheat phenotype, we deleted the descriptions about hypersensitive reactions. HR in Table S2 represent “homozygous resistance”, and we also added the legends in Table S2. The changes were highlighted in red color.
Line 89. What is ‘IT’, define it.
Answer: We modified the abbreviation of infection types (IT). The changes were highlighted in red color.
Line 90-91. In contrast, CS wheat was highly susceptible to powdery mildew with lesion area greater than 80%, and the IT grade was 4.
Answer: Thank you for reviewer’s modification, we revised the sentence. The changes were highlighted in red color.
Line 93. the infection type in SJ106? You mean ‘the types of responses to Bgt infection?
Answer: Thank you for reviewer’s modification, we used ‘the types of responses to Bgt infection’ in stead of ‘the infection type’. The changes were highlighted in red color.
Line 93-94. Get rid of ‘Thus, the powdery mildew resistance in SJ106 was controlled by the major gene locus.’
Answer: We got rid of that sentence.
Line 104. What is BSR-sequencing? define it, please.
Answer: BSR-sequencing was abbreviation to bulked segregant RNA-Seq. We added it in the introduction part. The changes were highlighted in green color.
Line 111. What is ‘the results of part 3.2,..’, nothing about this is mentioned in the test.
Answer: There was a mistake, we have got rid of it.
Line 116. SJ106 chromosome?
Answer: We added ‘chromosome’ in part 2.3. The changes were highlighted in red color.
Line 119. Spell out ‘SSR’
Answer: SSR was abbreviation to simple sequence repeat. We added it in the introduction part. The changes were highlighted in green color.
Line 120. amplify consistently the polymer…
Answer: We modified the sentence according the suggestions. The changes were highlighted in red color.
Line 122. A genetic linkage map..?
Answer: We modified the sentence according the suggestions. The changes were highlighted in red color.
Line 125. …between 4.90-8.97.., Is this a coordinate in genome?
Answer: We modified the sentence according the suggestions. The changes were highlighted in red color.
Line 125-128, rephrase the sentences. It looks like redundancy.
Answer: We modified the sentence as “Combined with 2.2 results, the candidate region was located on chromosome 6DS, the physical region was between 4.90-8.97 Mb. As none Pm genes reported in this region, PmSJ106 locus is likely to contain a novel Pm resistant gene.” The changes were highlighted in red color.
Line 128. Pm resistant genes?
Answer: We modified the sentence according the suggestions. The changes were highlighted in red color.
Line 131. PCR-Amplification with SSR markers in parents, resistant pool and F2 individuals.
Answer: We modified the sentence according the suggestions. The changes were highlighted in red color.
Line 133. …individual plants.
Answer: We modified the words according the suggestions. The changes were highlighted in red color.
Line 141. 59, Fifty-nine..
Answer: As we revised Line 142-143, word ‘59’ was deleted.
Line 142-143. ‘..were predicted to encode putative R proteins where 19 encoded NLRs and the remaining 40 encoded kinase proteins.’. I think there is a confusion here. Historically R-proteins were NLRs, however, you need to make sure not all kinase proteins are categorized as R-proteins. It would be better if you mention these as resistance-associated proteins?
Answer: Thank you for reviewer’s modification, we modified this sentence as “As reported Pm genes were always NLRs, transporter and kinase, such Pm21 and Pm13. We further annotated candidate genes in this region. Nineteen genes were predicted to encoded NLRs and forty encoded kinase proteins.” The changes were highlighted in red color.
Line 143. Fifteen of forty…
Answer: We modified the words according the suggestions. The changes were highlighted in red color.
Line 145-146. …we compared the gene sequences of the Chinese Spring genome (V2.1).
Answer: We modified the sentence according the suggestions. The changes were highlighted in red color.
Line 161. …a typical..
Answer: We modified the words according the suggestions. The changes were highlighted in red color.
Line 163. TaRGA5-like? Where is it originated? Should have a citation.
Answer: We added the explanations in Part 2.5. The changes were highlighted in red color.
Line 164-166. It is the coded protein sequences that are more important. Why no show a protein alignment?
Answer: We added figure S2 that showed a protein alignment. The changes were highlighted in red color.
Line 186. Agrobacterium tumefaciens; binary vector?
Answer: We added the explanations in Part 2.7. The changes were highlighted in red color.
Line 186-189. Tobacco leaves were infiltrated with Agrobacterium tumefaciens containing a binary vector carrying TaRGA5-like-GFP to transiently express and observe TaRGA5-like-GFP with a confocal fluorescence microscopy.
Answer: We modified the sentence according the suggestions. The changes were highlighted in red color.
Line 189. Is that a confocal fluorescence microscopy?
Answer: We observed GFP with a confocal fluorescence microscopy, but we just uses its fluorescence function. Therefore, we deleted word ‘confocal’ in the text. The changes were highlighted in red color.
Line 192. 35S: GFP, GFP vector control.
Answer: We modified the words according the suggestions. The changes were highlighted in red color.
Line 193. Get rid of ‘Green fluorescent protein (GFP) emitted green’. All color shown technically are pseudo color.
Answer: We got rid of it.
Line 196 and in Material and Methods. VIGS is not clearly described. What VIGS vector is used here?
Answer: We added the descriptions about VIGS in Material and Methods. The changes were highlighted in red color.
Line 200-202. The statement ‘These results indicated that TaRGA5-like silencing reduced powdery mildew resistance in SJ106 to some extent, but the decrease was not significant.’ conflict with the figure, which told the VIGS-silenced plants were significantly different. Fix it.
Answer: According to the reviewer’s suggestion, we modified the contents in 2.8 and 4.9, and modified a mistake in figure 9 and 11 by checking the figure file submitted on line. To clarify the VIGS and OE results, we added a Table S4. The changes were highlighted in red color.
Line 210. Remove ‘Meanwhile,’
Answer: We modified the words according the suggestions. The changes were highlighted in red color.
Line 231. Are ‘Cultivating and planting’ redundant?
Answer: We deleted ‘and planting’.
Line 236. What does ‘polymerization of multiple R genes’ mean? Diversification?
Answer: We modified as ‘R-gene polymerization’. The changes were highlighted in red color.
Line 253. only one, Pm45, was identified …?
Answer: We modified the words according the suggestions. The changes were highlighted in red color.
Line 259-260. Using the same unit ‘(19/4.07 Mb) vs (2,500/16 Gb)’.
Answer: ‘16 Gb’ was modified as ‘16000 Mb’. The changes were highlighted in red color.
Line 394. formal analysis? Are there any informal analysis?
Answer: We modified the contributions. The changes were highlighted in red color.

Reviewer 2 Report
Comments and Suggestions for Authors
In this manuscript, the authors identified a NLR gene, TaRGA5-like, from wheat SJ106, further analysis the function in powdery mildew resistance using transgenic line and VIGS plants.
TaRGA5-like play an important role in wheat resistance to powdery mildew fungi, SJ106 may become ideal doner for wheat in breeding varieties with durable, broad-spectrum disease resistance.
This manuscript is generally well described, I recommend accepting and publishing this MS in IJMS. However, there are several concerns the authors need to address.
1. English writing should be improved, e.g., Line66-69,70-72, with a little puzzling, please check through.
2. Line70 For wheat breeding?
3. Line71 ……functional study…… - functional analysis
4. Line72 transferring it to ---into
5. Line92 Bgh-Bgh should be italicized.
6. Line105-106 confirm the reads number.
7. Line231-242 here require additional references.
8. The authors described the LRR region sequence difference between the resistance individuals and susceptible individual, the TaRGA5-like gene expression profile also significantly different between CS and SJ106. So, it will be better, if the authors can describe the different in promoter region of TaRGA5-like gene between CS and SJ106.
9. The quality of Fig11 is too low.
10. Give a brief introduction of BSR-SEQ in introduction part.
Comments on the Quality of English Language1. English writing should be improved, e.g., Line66-69,70-72, with a little puzzling, please check through.
Author Response
Dear Reviewer,
Thanks very much for taking your time to review this manuscript. I really appreciate all your comments and suggestions! Please find my itemized responses in below and my revisions/corrections in the re-submitted files.
Sincerely,
Zhenying Wang
Appended to this letter is our point-by-point response to the comments raised by the reviewers. All changes in the text are highlighted in different colors.
- English writing should be improved, e.g., Line66-69,70-72, with a little puzzling, please check through.
Answer: We modified the sentences according the suggestions. The changes were highlighted in red color.
- Line70 For wheat breeding?
Answer: We modified the sentences according the suggestions. The changes were highlighted in green color.
- Line71 ……functional study…… - functional analysis
Answer: We modified the words according the suggestions. The changes were highlighted in green color.
- Line72 transferring it to ---into
Answer: We modified the words according the suggestions. The changes were highlighted in red color.
- Line92 Bgh-Bgh should be italicized.
Answer: We modified Bgt in italic. The changes were highlighted in red color.
- Line105-106 confirm the reads number.
Answer: We modified the raw reads and clean reads number. The changes were highlighted in green color.
- Line231-242 here require additional references.
Answer: We added two references. The changes were highlighted in green color.
- The authors described the LRR region sequence difference between the resistance individuals and susceptible individual, the TaRGA5-like gene expression profile also significantly different between CS and SJ106. So, it will be better, if the authors can describe the different in promoter region of TaRGA5-like gene between CS and SJ106.
Answer: We amplified the promoter region of TaRGA5-like, but there was no difference, data was not shown in this study.
- The quality of Fig11 is too low.
Answer: We modified the figure 11, and improve its resolution.
- Give a brief introduction of BSR-SEQ in introduction part.
Answer: We added the introduction of of BSR-Seq in introduction part. The changes were highlighted in green color.
